# High-Precision Inertial Sensor Charge Management Based on Ultraviolet Discharge: A Comprehensive Review

**DOI:** 10.3390/s23187794

**Published:** 2023-09-11

**Authors:** Tao Yu, Yuhua Wang, Yang Liu, Zhi Wang

**Affiliations:** 1Changchun Institute of Optics, Fine Mechanics and Physics, Chinese Academy of Sciences, Changchun 130033, China; wangyuhua21@mails.ucas.ac.cn (Y.W.); liuyang21h@mails.ucas.ac.cn (Y.L.); 2School of Electronic Information Engineering, Changchun University of Science and Technology, Changchun 130022, China; 3University of Chinese Academy of Sciences, Beijing 100049, China; 4School of Fundamental Physics and Mathematical Sciences, Hangzhou Institute for Advanced Study, University of Chinese Academy of Sciences, Hangzhou 310024, China

**Keywords:** space gravitational wave detection, inertial sensor, charge accumulation, charge management, ultraviolet discharge

## Abstract

The charge accumulation caused by cosmic rays and solar energetic particles poses a significant challenge as a source of noise for inertial sensors used in space gravitational wave detection. To address this issue, the implementation of charge management systems based on ultraviolet discharge becomes crucial. This paper focuses on elucidating the principles and methods of using ultraviolet discharge for charge management in high-precision inertial sensors. Furthermore, it presents the design and implementation of relevant payloads. Through an analysis of the charge accumulation effect and its impact on noise, key considerations regarding coatings, light sources, and optical paths are explored, and some current and valuable insights into the future development of charge management systems are also summarized. The conclusions drawn from this research also provide guidance for the advancement of higher precision ultraviolet discharge technology and the design of charge management systems.

## 1. Introduction

The inertial sensor, consisting of a test mass (TM), an electrode housing (EH), and associated peripherals, plays a crucial role in space missions [1,2,3,4]. In space gravitational wave detection missions, inertial sensors are used not only as inertial references to provide measurement data for the drag-free control system [5,6] but also as the end mirrors of the high-precision inter-satellite laser interferometer to capture the gravitational wave signal by measuring the change in distance between two free-floating TMs located inside different spacecraft [7,8,9]. Strict control over non-conservative force interferences on the TMs is therefore essential. One of the primary sources of noise that is challenging to eliminate is cosmic rays and high-energy particles [10,11]. When these particles or their secondary counterparts interact with the TMs, they cause charge accumulation [12]. These accumulated charges then interact with the electromagnetic field in the environment where the TMs are located, thus causing related acceleration noise [13]. Therefore, in precision space missions, a low-disturbance charge management system (CMS) must be employed to effectively control the accumulated charges on the TMs.

At present, the charge management methods for space-isolated conductors mainly include conductive gold wire, a conductive film, gas glow discharge, an electron/ion beam, and ultraviolet (UV) discharge [14,15]. In early space missions such as CHAMP (Challenging Minisatellite Payload) [16,17], GRACE (Gravity Recovery and Climate Experiment) [18,19], GOCE (Gravity Field and Steady-State Ocean Circulation Explorer Mission) [20,21] and MICROSCOPE (Micro-Satellite with Drag Control for the Observation of the Equivalence Principle) [22,23], where higher precision inertial sensors are not required, the conductive gold wire, which uses μm-diameter gold wires to connect a TM and an EH for charge management, is an effective solution. It has a compact structure and is easy to implement, but it also introduces additional acceleration noise due to the physical connection [24]. Existing calculations indicate that this noise is on the order of $10^{-13}$ m/s${}^{2}$/Hz${}^{1/2}$ [25], which severely limits the sensitivity of inertial sensors.

With the increasing demand for the accuracy of inertial sensors in space missions, the traditional conductive gold wire has become inadequate to meet the higher requirements. In response to this challenge, the development of UV discharge based on the photoelectric effect has gained prominence [26,27]. UV discharge uses specific wavelengths of UV light to irradiate the gold-coated TMs, which helps to release photoelectrons and then controls the movement direction of photoelectrons by the local electric fields, thereby removing the residual charges on the TMs without contact [28]. Compared to conductive gold wire, UV discharge achieves charge management without introducing large interference noise. It is a low-interference charge management method. Since the successful space verification of the GP-B (Gravity Probe B) mission [29,30], UV discharge has garnered significant attention in related fields. It is now recognized as the most promising charge management technique for future high-precision inertial sensors of space missions [31].

Aiming at the charge management of high-precision inertial sensors in space gravitational wave detection missions, this paper briefly introduces the charge accumulation on the TMs and the resulting noise. Based on this, the principles and methods of charge management using UV discharge are described, as well as the design and implementation of related payloads. Finally, the key issues related to coating, light source, and optical path in the future development of CMSs are analyzed and prospected. These results provide guidance for the advancement of higher precision UV discharge technology and the design of CMSs.

## 2. The Charge Accumulation of Inertial Sensors

### 2.1. Inertial Sensor for Space Gravitational Wave Detector and Its Charge Accumulation

The inertial sensor is a scientific instrument used to measure acceleration in microgravity environments. Its core structure consists of a sensitive probe, a capacitive (or optical) displacement sensor, and a feedback control circuit [32,33]. Based on its principle, it can operate in two different modes during its on-orbit functionality: accelerometer mode and inertial reference mode [25,34]. In the accelerometer mode, the TM is suspended within the EH, which remains fixed within the spacecraft. As the TM and EH experience relative motion, the sensing circuit on the EH detects the displacement and actuates the voltage on the plate through the controller. This generates an electrostatic force that pulls the TM back to its central position. This closed-loop operation allows the inertial sensor to perform applications such as satellite gravity measurement and equivalence principle testing. On the other hand, in the inertial reference mode, the TM serves as an inertial reference body. When the sensing circuit detects the relative motion between the TM and the EH, the satellite’s micro-thruster generates a compensating thrust. This allows the spacecraft to follow the motion of the TM, improving its microgravity and vibration levels (known as drag-free control) [32,35]. Gravity gradient satellites often benefit from this mode [36]. In future space gravitational wave detection missions, their inertial sensors will operate in the inertial reference mode for sensitive axes (scientific measurement axes) and the accelerometer mode for non-sensitive axes simultaneously, accommodating different degrees of freedom and frequency bands [37,38].

According to the requirements of space gravitational wave detection missions, the sensitive probe of their inertial sensors consists of a cube TM and EH [39]. Figure 1 shows the inertial sensors used in LISA (Laser Interferometer Space Antenna). The TM is a 2 kg cube with a side length of about 46 mm, made of 75% gold and 25% platinum for low magnetic susceptibility and residual magnetic moment to avoid magnetic effects [40]. The EH, larger than the TM, is a cubic shell made of molybdenum with a gap of 3–4 mm to the TM [41]. To enable capacitive sensing and actuation, eighteen molybdenum electrodes are distributed over six surfaces of the EH (Figure 1b): twelve sensing/actuation electrodes (four along each *x*, *y*, and *z* axis) and six injection electrodes (four along the *z*-axis and two along the *y*-axis). These electrodes are insulated from the EH by sapphire substrates [42].

During operation, as shown in Figure 2, the position and attitude of the TM within the EH are sensed and controlled by a six-degree-of-freedom differential capacitance bridge composed of six pairs of differential capacitances (consisting of 12 sensing/actuation electrodes and TM, such as $C_{P1}$ and $C_{P2}$) and corresponding inductances (such as $L_{1}$ and $L_{2}$) [40]. When sensing, a 100 kHz injection signal is applied to the TM via the six injection electrodes. When the TM deviates from its central position, the differential capacitance signal is generated in the differential capacitance bridge. After a series of transformations, the sensor outputs the corresponding voltage [43]. During control, the drag-free control system (DFACS) uses the measurement data to calculate the required low-frequency (60–270 Hz to prevent interference with the injection signal) sinusoidal electrostatic actuation voltage and applies it to the transformer bridge, generating electrostatic forces on the TM to keep it centered within the EH along the non-sensitive axes [44]. In the sensitive axis direction (i.e., *x*-axis), DFACS will control the operation of micro-thrusters to make the satellite follow the movement of the TM based on the measurement data, to compensate for the influence of external disturbances on the satellite’s movement and to provide an ultra-quiet and ultra-stable platform for precision experiments [45].

Once deployed in orbit, inertial sensors are inevitably affected by charge accumulation, which arises from solar energetic particles and cosmic rays, as well as the charge transferred by physical contact when the TM is released [46,47]. Among these, solar energetic particles and cosmic rays have a greater effect, while the flux and energy spectrum of the charged particles captured by the spacecraft depend on its orbit [48,49]. There are two mechanisms by which these charged particles promote TM charging [50]: one is that the charged particles accumulate directly on the TM (direct charging); the other is that the charged particles interact with the internal material of the spacecraft to produce secondary particles which then accumulate on the TM (indirect charging). Research indicates that indirect charging has a more pronounced effect on TM charging [51].

The influence of charge accumulation on inertial sensors depends not only on the particles themselves but also on the specifics of the space mission [52]. Generally, particles hitting the TM impart momentum and cause a temperature elevation. Additionally, they alter the coupling between the TM and the spacecraft [53] and can introduce coherent Fourier components into scientific data [54,55]. However, for the space gravitational wave detection missions, the primary concern remains the acceleration noise resulting from the interaction between accumulated charges and the surrounding electromagnetic field environment. This noise is estimated to limit the sensitivity of inertial sensors, especially in the frequency band below 0.1 mHz [56,57].

### 2.2. Charge-Induced Acceleration Noise

When the inertial sensor is in orbit, the amount of charge accumulated on the TM varies according to the average charging rate [58]. Assuming isotropic charging events, the entire charging process can be viewed as the accumulation of multiple independent charging events with a net charge of qe (where q=+1 for protons). Hence, the amplitude spectral density of the current noise caused by the shot noise can be expressed as follows: (1)$$S_{q}^{1/2}=\sqrt{2eI}$$
where *e* is the elementary charge, and the equivalent current *I* can be replaced by the effective charging rate $\lambda_{eff}$ (representing the probability of a single charge hitting the TM event within 1 s) [59], thus: (2)$$S_{q}^{1/2}=\sqrt{2e\left(\lambda_{eff}e\right)}=e\sqrt{2\lambda_{eff}}$$

The accumulated charge, *Q*, is the result of integrating the current. Thus, the amplitude spectral density of the random charging noise can be obtained by integrating Equation (Equation 2) in the time domain. In the frequency domain, we have (*f* is the frequency) [60,61]: (3)$$S_{Q}^{1/2}=\frac{1}{2\pi f}S_{q}^{1/2}=\frac{e}{2\pi f}\sqrt{2\lambda_{eff}}$$

The charge accumulated in the random charging process will generate an electrostatic force with the electric field environment in which the TM is located. This part is the dominant part of the noise influence related to charge accumulation [62,63], so it is also the main part of the CMS to be solved.

In the simplified electrostatic model of the inertial sensor, the electric field energy of the capacitance system composed of the TM and the EH is: (4)$$E=\frac{\sum_{i}C_{i}{\left(V_{i}-V_{tm}\right)}^{2}}{2}$$
where $C_{i}$ is the capacitance of the capacitor composed of the electrode *i* and the TM, $V_{i}$ is the voltage applied to electrode *i*, and $V_{tm}$ is the potential of the TM, and: (5)$$V_{tm}=\frac{Q}{C_{T}}+\frac{1}{C_{T}}\sum_{i}C_{i}V_{i}$$
where *Q* is the amount of charge accumulated on the TM, and $C_{T}$ is the total capacitance of the capacitor system.

The electrostatic force acting on the TM at the sensitive axis can be obtained by the partial derivative of the electric field energy in the *x*-axis direction, and the corresponding acceleration is: (6)$$a_{Ex}=\frac{1}{2m}\sum_{i}\frac{\partial C_{i}}{\partial x}V_{i}^{2}+\frac{Q^{2}}{2mC_{T}^{2}}\frac{\partial C_{T}}{\partial x}-\frac{Q}{{mC}_{T}}\sum_{i}V_{i}\frac{\partial C_{i}}{\partial x}$$
where *m* is the mass of the TM, the first term is the acceleration corresponding to the force providing electrostatic control, which is independent of the charge accumulation; the second term is the acceleration due to the change in total capacitance caused by the random charging; and the third term is the acceleration due to the applied voltage on the plate and the electrostatic force between the stray potential and the TM after charging.

When the TM is at the central position, if the area of the four electrodes in the *x*-axis direction is *A*, the distance between the electrodes and the TM is *d*, and the dielectric constant of the gap is $\varepsilon_{0}$, then the nominal capacitance is $C_{x}=\varepsilon_{0}\frac{A}{d}$. When there is a small offset *x* between the TM and its central position, $C_{i}=\varepsilon_{0}\frac{A}{d\pm x}=C_{x}\frac{1}{1\pm \frac{x}{d}}$, so the capacitance gradient along the *x*-axis direction is: (7)$$\frac{\partial C_{i}}{\partial x}=\frac{\partial }{\partial x}\left(C_{x}\frac{1}{1\pm \frac{x}{d}}\right)=C_{x}\frac{\mp \frac{1}{d}}{{\left(1\pm \frac{x}{d}\right)}^{2}}\approx \mp \frac{C_{x}}{d}$$

In fact, the TM only moves freely along the *x*-axis (the *y*-axis and *z*-axis are electrostatically controlled). At this time, the change in $C_{T}$ can simply be considered as the change in two pairs of capacitances on the *x*-axis, that is: (8)$$\frac{\partial C_{T}}{\partial x}=\frac{\partial C_{1}}{\partial x}+\frac{\partial C_{2}}{\partial x}+\frac{\partial C_{3}}{\partial x}+\frac{\partial C_{4}}{\partial x}={2C}_{x}\frac{\frac{4x}{d^{2}}}{{\left[1-{\left(\frac{x}{d}\right)}^{2}\right]}^{2}}\approx \frac{8C_{x}x}{d^{2}}$$

Therefore, the electrostatic acceleration noise caused by random charging can be expressed as: (9)$$S_{a_{E}}^{1/2}=\left(\frac{8QC_{x}x}{mC_{T}^{2}d^{2}}+\frac{8V_{c}C_{x}x}{mC_{T}d^{2}}+\frac{\Delta xC_{x}}{mC_{T}d}\right)S_{Q}^{1/2}$$

In Equation (Equation 9), the first term corresponds to the second term in Equation (Equation 6), and the last two terms correspond to the third term in Equation (Equation 6), which represents the acceleration noise generated by the common mode component $V_{c}$ in $V_{i}$ on the electrode and the charge fluctuation when the TM deviates from the center, and the acceleration noise generated by the differential mode voltage and the charge fluctuation on the electrode when the TM is at the center. The differential mode component in $V_{i}$ can be generated by the applied voltage or the stray potential. To simplify the analysis, only the stray potential ${\delta V}_{i}$ is considered here, that is, $\Delta x=\frac{\sum_{i}\frac{\partial C_{i}}{\partial x}{\delta V}_{i}}{\left|\frac{\partial C_{i}}{\partial x}\right|}$.

In addition to the electrostatic force, the accumulated charge on the TM will interact with the internal and external magnetic fields of the spacecraft to generate a Lorentz force [64]. However, compared to the electrostatic force, this Lorentz force has a much smaller influence and can be well suppressed by magnetic shielding measures [65]. Its corresponding acceleration is [61]: (10)$$a_{Lx}=\frac{Q{\vec{v}}_{SC}{\vec{B}}_{ext}}{m}+\frac{{Q\vec{v}}_{TM}{\vec{B}}_{ext}}{m}+\frac{{Q\vec{v}}_{TM}{\vec{B}}_{int}}{m}$$
where ${\vec{v}}_{SC}$ is the velocity of the spacecraft, ${\vec{v}}_{TM}$ is the velocity of the TM relative to the spacecraft, ${\vec{B}}_{ext}$ is the magnetic induction intensity of the interplanetary magnetic field, and ${\vec{B}}_{int}$ is the magnetic induction intensity of the magnetic field in the spacecraft. The acceleration noise associated with random charging is [62]: (11)$$S_{a_{L1}}^{1/2}=\frac{1}{m}\left({\vec{v}}_{SC}{\vec{B}}_{ext}+{\vec{v}}_{TM}{\vec{B}}_{ext}+{\vec{v}}_{TM}{\vec{B}}_{int}\right)S_{Q}^{1/2}$$

### 2.3. Charge Management Requirements for Space Gravitational Wave Detection Mission

In space gravitational wave detection missions, it is crucial to address the technical challenges surrounding high-precision inertial sensors, and the acceleration noise arising from the interaction between the accumulated charges and the surface stray potential most need to be suppressed [58,66]. The LISA mission, which represents the most mature space gravitational wave detection plan, sets an upper limit on the amplitude spectral density of the acceleration noise caused by TM charging at its operating frequency, as described in Equation (Equation 12) [67,68].
(12)$$S_{a\;\mathrm{charge}}^{1/2}\left(\omega \right)\approx 0.8\times 10^{-15}\frac{m}{s^{2}\sqrt{\mathrm{Hz}}}\times \left(\frac{4\;\mathrm{mm}}{gap}\right)\left(\frac{V_{dc}}{10\;\mathrm{mV}}\right){\left(\frac{\lambda_{\mathrm{eff}\;}}{300\;s^{-1}}\right)}^{1/2}\left(\frac{0.1\;\mathrm{mHz}}{f}\right)$$

To meet this requirement, two measures must be taken. First, it is essential to minimize the stray potential on all internal surfaces of the inertial sensor. Second, a CMS must be incorporated to reduce the cumulative net charge on the TM (with an upper limit of $10^{-12}$ C in LISA) [69]. Given the extremely low noise tolerances in scientific measurements for space gravitational wave detection missions, non-contact discharge technology using UV discharge based on the photoelectric effect emerges as the optimal choice [70,71]. However, it is important to note that the discharging process realized by CMSs also independently generates random charging noise [72]. Therefore, Equation (Equation 3) should actually be: (13)$$S_{Q}^{1/2}=\frac{e}{2\pi f}\sqrt{2\left(\lambda_{eff}+\lambda_{CMS}\right)}$$
where $\lambda_{CMS}$ is the equivalent effective charging rate of the CMS. Therefore, the dynamic range of the discharge rate must be strictly limited to balance the noise caused by the discharge process of the CMS [70]. In order to meet the requirements of the two modes of operation, rapid discharge after TM release and continuous discharge during scientific measurement, the dynamic range of the discharge must be considered from the following two points [73]:Rapid discharge mode: Discharging the TM potential of ±1 V within 1 h requires a maximum discharge rate of $\pm 6\times 10^{4}$ e/s (where the total capacitance is 34.2 pF).Continuous discharge mode: To counterbalance the anticipated charging effects induced by the on-orbit space environment and to allow for margins, the minimum discharge rate must meet ±10 e/s.

In summary, the dynamic range required for positive and negative discharge rates within the CMS is 10 e/s–$6\times 10^{4}$ e/s, and in order to achieve accurate charge control, the discharge rate adjustment within this range needs to be accomplished in steps of ±10 e/s. Therefore, the final discharge rate of the CMS needs to be adjusted by at least $1.2\times 10^{4}$ steps [73].

## 3. Charge Management Based on UV Discharge

### 3.1. Principle of UV Discharge

When UV light is irradiated onto a metal surface, if the photon energy hv of the absorbed light exceeds the work function ϕ of the metal, valence electrons in the metal are excited and rise above the vacuum level, transforming into free electrons [74,75]. This phenomenon follows the principles of the photoelectric effect, where the kinetic energy *E* of the emitted electrons can be described by the equation: (14)E=hv−ϕ

According to existing research [76,77], at absolute zero (i.e., ignoring the effect of temperature on photoelectron emission), the equivalent photocurrent *I* generated by a large amount of photoelectron emission is: (15)$$I\approx A{\left(hv-\phi -eV\right)}^{2}$$
where *A* is a constant, and *V* is the local bias voltage between the emitting and receiving surfaces, which can be used to promote or inhibit the transfer of photoelectrons between the two surfaces. When *V* is positive, only photoelectrons with kinetic energy components higher than eV in the direction perpendicular to the emitting surface are likely to be transferred [78], and whether these photoelectrons can eventually contribute to the TM discharge depends mainly on their trajectories between the emitting and receiving surfaces (the trajectories are related to the initial position, velocity direction, and kinetic energy of the photoelectrons) [79].

In the process of UV discharge, UV light is inevitably reflected, so that both the emitting and receiving surfaces generate a photocurrent for discharge. The polarity of the net photocurrent at this stage determines the discharge polarity, while the magnitude of the net current determines the discharge rate. The former depends primarily on the local bias voltage polarity and the irradiation mode, while the latter depends on the local bias voltage magnitude, the quantum yield, and the absorption coefficient of the emitting surface. Among these factors, the irradiation mode is typically fixed, and the variables related to surface material properties, such as the quantum yield and absorption coefficient, are difficult to control effectively [80]. Therefore, adjusting the polarity and magnitude of the local bias voltage is the key to realizing the control of the discharge polarity and rate [81]. However, it should be noted that this control ability of bias voltage is also limited because as the discharge progresses (as shown in the parallel plate case in Figure 3, where the yellow lines represent the photocurrent), the transferred photoelectron will gradually generate an internal electric field, so that the photocurrent emitted from the two surfaces tends to be balanced, reaching the so-called charge balance state [82].

### 3.2. Charge Management Method Based on UV Discharge

Based on the above discharge principle, the DC discharge method shown in Figure 4 was first introduced in the GP-B mission [26]. This method involved applying a suitable bias voltage to the EH electrodes (there are two dedicated CMS electrodes on the EH in GP-B) while the UV light source, whose purpose was to control the relationship between the photoelectron currents $I_{TM}$ on the TM and $I_{EH}$ on the EH electrodes, was continuously open. Specifically, with the plate as the zero potential, when the TM is positively biased, $I_{TM}<I_{EH}$, and the TM potential increases, whereas when the TM is negatively biased, $I_{TM}>I_{EH}$, and the TM potential decreases; the greater the bias voltage, the greater the net photocurrent, and the faster the TM potential changes.

The DC discharge is easy to implement, and the charge management with an adjustable bipolar rate can be realized by controlling the polarity and magnitude of the bias voltage. However, in more sophisticated missions such as LISA and LPF (LISA Pathfinder), the introduction of a dedicated bias voltage also means introducing more noise, which can significantly affect scientific measurements. To effectively address this problem, it would be beneficial to reuse the existing periodic injection signal in the inertial sensor [83]. Therefore, the AC discharge method shown in Figure 5 has been proposed. In this method, the polarity and magnitude of the bias voltage are no longer controlled directly, but by controlling the operating state of the UV lamp to synchronize with the injection voltage signal, so that the UV lamp is turned on only when the local bias voltage can promote the electron flow in a certain expected direction and suppress the undesired electron flow [84]. Specifically, when the two are in phase (0${}^{\circ }$), $I_{TM}>I_{EH}$, and the TM potential increases. If they are out of phase (180${}^{\circ }$), $I_{TM}<I_{EH}$, and the TM potential decreases.

Essentially, the two discharge methods above both use local electric fields to regulate the overall flow of photocurrent to achieve charge management. In contrast, the AC discharge method does not require a dedicated bias voltage signal. Instead, it cleverly uses the existing injection voltage signal by manipulating the operating state of the UV lamp (requiring faster response light sources, such as UV LEDs). As a result, it offers lower noise, greater adaptability, and enables the implementation of more intricate control strategies. Nevertheless, the accuracy and effectiveness of the discharge, whether DC or AC, are highly dependent on an understanding of the photoelectric properties of the surface. Consequently, calibration becomes a tedious task, and, as the physical parameters change over time, accuracy can be significantly affected. This poses a clear risk to long-term on-orbit missions.

In response to the aforementioned challenges, Yang F. et al. [85] introduced an adaptive discharge method in 2020 based on a differential illumination model, as shown in Figure 6. This method allows the discharge polarity and rate to be adjusted by changing the relative optical power of two UV lamps that illuminate the respective surfaces simultaneously. By ensuring that the optical power ratio closely matches the required compensation coefficient for discharge, complex physical parameter measurements are unnecessary. Essentially, this method compensates for changes in complex physical parameters by calibrating the compensation coefficient. However, it’s also sensitive to variations in photoemission parameters and light intensity, so periodic recalibration may be required, but the specific implementation of this method involves calibrating numerous compensation coefficients, which can be quite intricate. To address this issue, Pi X. et al. [86] proposed a convenient calibration method based on linear estimation in 2023, that can greatly improve the efficiency of the compensation coefficient calibration, thereby reducing the difficulty of the adaptive discharge method implementation and making it more feasible. However, in general, the technology still needs to be verified in real orbit.

The use of shorter wavelength UV light is required in the adaptive discharge method to excite photoelectrons with higher energy in order to maximize the photoelectric quantum efficiency and thus maximize the generated photocurrent. However, due to the requirement of low additive noise in the charge management process, the optical power output of the UV lamp needs to be limited to achieve a low photocurrent discharge process. In addition, the complicated calibration is also a great challenge. In this regard, S. Wang et al. [87] proposed a slow photoelectron method in 2022, which uses the lower energy photoelectrons possible (longer UV wavelength, 275–300 nm). In this method, slow photoelectrons will act as a low-resistance short-circuit between the TM and the EH when the TM is connected to a variable bias voltage, which is insensitive to the system conditions (vacuum, temperature) or photoemission parameters (quantum efficiency, reflectance) and light intensity variation. This allows for better utilization of the existing high-power UV lamps and meeting the low-current charge management requirement. However, it was developed for MGRS-type sensors, i.e., there is no electric field between the TM and the EH, so if it is to be applied to inertial sensors like LISA, the problem of local electric field interference must be addressed.

## 4. CMS Based on UV Discharge

### 4.1. Development of CMSs Based on UV Discharge

Up to now, several institutions and research teams, including the University of Trento [88,89], University of Washington [90], Imperial College [46,91], Stanford University [87,92,93], University of Florida [94,95,96], Huazhong University of Science and Technology [97,98,99], and Lanzhou Institute of Space Technology Physics [100] have published significant results from their ground UV discharge experiments. These studies have introduced many innovative ideas and methods for charge management, providing valuable experience for the design and development of future CMSs. Based on these ground research results, the UV discharge technology has been successfully applied in the CMS for space missions such as GP-B and LPF.

#### 4.1.1. CMS of GP-B

The GP-B mission, shown in Figure 7a, aims to study the geodesic effect and the frame-dragging effect by measuring the precession of the Earth using a suspended gyroscope in a polar orbit at an altitude of 650 km to verify the theory of general relativity [101]. Its core payload is four quartz gyroscopes (TM) with niobium film encapsulated in a Dewar bottle, and an EH with three pairs of orthogonal electrodes on its periphery as shown in Figure 7b [29]. To meet gyroscope torque requirements and prevent charge accumulation leading to de-suspension, it is crucial to maintain the potential of the TM below 15 mV [11]. Therefore, a CMS capable of achieving DC discharge was developed to provide non-contact bipolar charge control of the TM.

The dedicated electrode and UV irradiation settings for charge management in GP-B are shown in Figure 8a. Its UV light source is a mercury lamp with a nominal light output of 10 μW and a peak wavelength of 254 nm. The UV light is transmitted to the TM via an optical fiber, while the polarity and rate of discharge are adjusted by controlling the polarity and amplitude of the bias voltage at the charge control electrode. The final discharge efficiency ranges from 20 to 200 fA/μW [29], and the UV discharge rate can reach $10^{5}$ e/s after the DC bias is applied. Figure 8b shows part of its on-orbit discharge results. When the UV lamp is turned on, the bias voltage is configured at ±3 V to achieve system equilibrium [102], and the final charge can be controlled below 5 mV [14,29].

#### 4.1.2. CMS of LPF

LPF is the key verification mission for the LISA mission [103]. It verified the core technologies such as inertial sensors, laser interferometry, and micro-thrusters that will be used in the future LISA mission [104]. The primary payloads of LPF are two inertial sensors, as shown in Figure 9. Similar to the charge management requirement of LISA, this mission necessitates that the accumulated charges on the TM remain below $10^{-12}$ C [105]. Adherence to this requirement is crucial to maintaining low noise interference. To achieve this, the LPF is equipped with a CMS that is capable of DC discharge and offers two discharge modes: rapid discharge and continuous discharge [69,106].

The CMS of LPF is shown in Figure 10, which consists mainly of the following three parts [107]:UV Light Unit (ULU): The ULU consists of six programmable mercury lamps, along with the necessary electronic equipment. To protect the optical element and prevent visible light from entering the inertial sensor, the emitted light from the mercury lamps is filtered by an optical tube and then focused into an optical fiber. Therefore, only UV light with a wavelength of 253.7 nm is used for discharge. Additionally, each lamp housing includes a thermistor, an ohmic heater, and a silicon carbide photodiode to monitor the status of the UV lamps.Fiber Optic Harness (FOH): Due to the limitations of the satellite arrangement, the FOH for UV light transmission is composed of 19 optical fibers with a diameter of 200 μm. Each lamp is independently connected to transmit light from the ULU located in the outer cabin of the spacecraft to each inertial sensor.Inertial Sensor UV Kit (ISUK): The UV light from the fiber is irradiated to the TM and EH at an angle of about 20° through an optical fiber with a diameter of 1 mm and a length of 75 mm. Since a large amount of light will be lost in the groove when the EH is irradiated, in order to reduce the effect of discharge asymmetry, each inertial sensor of LPF has three UV kits, two of which point at the EH and one point at the TM.

After on-orbit adjustment and calibration, the LPF CMS performed a series of on-orbit experiments from March 2016 to July 2017. During this period, six mercury lamps were lit a total of 418 times, resulting in 421 h of trouble-free operation. The obtained results indicate that the average quantum yield of the irradiated TM is $\left(2.6\pm 0.3\right)\times 10^{-5}$ e/photon, while the average quantum yield of the irradiated EH is $\left(1.2\pm 0.2\right)\times 10^{-5}$ e/photon. Figure 11 shows its on-orbit discharge curves [107]:

#### 4.1.3. CMS of SaudiSat-4

Limited by technology and devices, the UV light sources used in GP-B and LPF are mercury lamps. These light sources are plagued by several issues, including slow response, limited dynamic range, temperature sensitivity, and short lifespan. Additionally, they serve as significant sources of radio frequency interference and electromagnetic interference, which are inconveniences for future space applications [108]. Fortunately, the emergence of commercial UV LEDs has provided a viable solution. These innovative light sources offer exceptional performance properties, making them the most ideal choice for the development of future CMSs [109]. However, it is crucial to evaluate their spatial feasibility. To address this need, Stanford University developed a UV LED-based discharge experimental payload, shown in Figure 12, and successfully completed the on-orbit experiment on the SaudiSat-4 mission launched in 2014, to verify the reliability and effectiveness of the space application of the UV LED-based CMS [28].

In the UV LED-based CMS developed by Stanford University, both the UV LED and the bias voltage are driven by 100Hz square wave signals with an adjustable duty cycle, and a probe is used to monitor the accumulated change on the TM. Some of the results are shown in Figure 13. The maximum discharge rate measured in orbit can reach $1.75\times 10^{8}$ e/s. Remarkably, no noticeable attenuation of the UV LEDs was observed during the entire mission [93].

### 4.2. CMS for Future Space Gravitational Wave Detection

As a key technical verification of the LISA CMS, the LPF CMS has exceeded expectations in its on-orbit validation [110,111], significantly increasing confidence in the use of UV discharge technology for high-precision inertial sensor charge management. However, facing the higher requirements of the future space gravitational wave detection mission, there are still many problems to be solved in the development of the related CMS in the measurement and recovery of the photoelectric properties of the coating, the packaging and control of the light source, and the design and arrangement of the optical path.

For the UV discharge process, the photoelectric properties of the coating (such as work function, quantum yield, reflectivity, etc.) are complex and critical issues [112]. The same coating will exhibit significant variations in different photoelectric properties, which can have an unpredictable impact on the final charge management effect [113]. In the inertial sensor used for space gravitational wave detection, in order to make the TM have high reflectivity in the infrared band to achieve laser interferometry and minimize the patch effect, almost all surfaces are coated with gold, except for the surfaces related to the locking and releasing system of the fixed TM during the launch [15]. Therefore, it is necessary to measure and analyze the photoelectric properties of the gold coating, including its dynamic changes, and find a method to maintain and recover these properties [80]. The LISA team has accumulated valuable research experience in analyzing the photoelectric properties of the coating and its dynamic changes. By optimizing the manufacturing process and storage environment of the gold coating, together with techniques like argon ion flow washing and high-temperature baking, partial recovery of the photoelectric properties of the inertial sensor surface was successfully achieved in the LPF mission [107]. In the future, more research is needed to minimize risks arising from these highly variable factors.

The packaging and control of the light source are other crucial factors in the UV discharge process. Generally, the work function of a vacuum-deposited gold coating is about 5.3 eV [114], but surface contamination often reduces this to about 4.6 eV [115]. Equation (Equation 14) indicates that only when the wavelength of the UV light source is in the appropriate range, can the gold-coated surface of inertial sensors emit photoelectrons [87]. In recent years, several commercial UV LEDs that meet the aforementioned requirements have been introduced successively [116]. Table 1 shows that UV LEDs offer significant advantages in energy consumption, packaging, stability, lifetime, and high-frequency modulation performance compared to traditional mercury lamp light sources. However, before formal application, thorough research must be conducted on their packaging and control technology to ensure their resilience under environmental stresses such as radiation, vibration, and impact [117,118,119]. Moreover, a deep understanding of the operating conditions and usage methods is necessary [120]. Presently, various studies have examined the feasibility of UV LED space applications [117,121,122] and UV LED control technology [100,123]. These results have provided a good foundation for the future practical implementation of CMSs based on UV LEDs.

In addition to the two factors mentioned above, the design of the UV light path is also a noteworthy issue in the development of future CMSs. In practical applications, it is crucial to minimize the thermal effect on the inertial sensor by locating the ULU, which consists of the lamp and its driver, as far away from the sensor as possible. Therefore, the UV light should be transmitted via optical fibers [96]. The selection and arrangement of the fibers, the coupling between the fiber and the lamp, and the position and angle of the final light outlet are all important considerations in the design process.

When selecting an optical fiber, the primary consideration is its transmission efficiency and anti-ultraviolet loss capability. Therefore, the anti-ultraviolet multimode fiber with a large numerical aperture and core diameter should be selected as much as possible [124]. However, due to the difficulty of bending the large core diameter fiber itself (metal coatings allow for reduced bend radius), the size of the core diameter is not the bigger the better and should be weighed against its arrangement carefully.When coupling optical fibers with UV lamps, the coupling efficiency is crucial. Other factors such as power consumption, thermal noise, coupling reliability, lifetime, and miniaturization also need to be taken into account. Generally speaking, when using a UV LED light source with a small divergence angle, lens coupling is a more suitable method compared to direct coupling and parabolic lens combination coupling [125]. However, during the actual coupling alignment process, it is important to ensure the co-axiality and alignment of the light source, lens, and fiber to minimize losses due to area adaptation, numerical aperture, and reflection.Finally, the design and optimization of the illumination mode of the light outlet should also be emphasized. The incident position determines the proportion of controllable photocurrent (only the photocurrent in the area affected by the electrodes can be controlled), and the incident angle affects the propagation path of UV light inside the inertial sensor, thus influencing the absorption coefficient of UV light on different surfaces. Both of them have a great influence on the discharge efficiency. Some proposals have been made to adjust the optical path by installing a transmitting mirror in the inertial sensor or cutting the fiber section to change the incident angle [126]. However, the feasibility of these ideas still needs to be assessed from an engineering perspective.

## 5. Conclusions

In this paper, the charge management of high-precision inertial sensors based on UV discharge is described from the noise influence of charge accumulation, UV discharge principle, and related CMS design. The main contents are summarized as follows:Inertial sensors are crucial payloads that provide inertial references for precision space missions. However, the charge accumulation caused by the space environment will inevitably affect their on-orbit performance. To mitigate this issue, it is essential to use a special CMS to reduce the residual charges on the TM and effectively suppress the associated noise. In order to meet the extremely low noise requirements for scientific measurements in space gravitational wave detection, the charge management of its high-precision inertial sensor requires the use of non-contact UV discharge technology. Nevertheless, the related CMS must also meet dynamic range constraints to strike a balance between the charge management requirements with the indirect noise effects resulting from the discharge process.The UV discharge uses the photoelectric effect to remove the residual charges on the TM. In order to achieve accurate control of the discharge polarity and rate, it is crucial to adjust the polarity and magnitude of the local bias voltage, considering the complexity of the physical process involved. Currently, there are two widely used charge management methods: DC discharge and AC discharge. The latter has more promising applications as it does not require a special bias voltage. However, its accuracy and effectiveness still largely depend on understanding the photoelectric properties of the surface coating. An adaptive discharge method, which calibrates the compensation coefficient to account for changes in complex physical parameters, can effectively address these issues. Nevertheless, its effectiveness still needs to be verified in actual scenarios.Currently, due to extensive research on the ground UV discharge, the non-contact CMS has been successfully implemented in various space missions such as GP-B, LPF, and SaudiSat-4. However, with the increasing demands of future space gravitational wave detection missions, there are still numerous challenges that need to be addressed in the development of CMS. These include measuring and recovering the photoelectric properties of the coating, packaging, and controlling the light source, as well as designing and arranging the optical path.

## Figures and Tables

**Figure 1 sensors-23-07794-f001:**
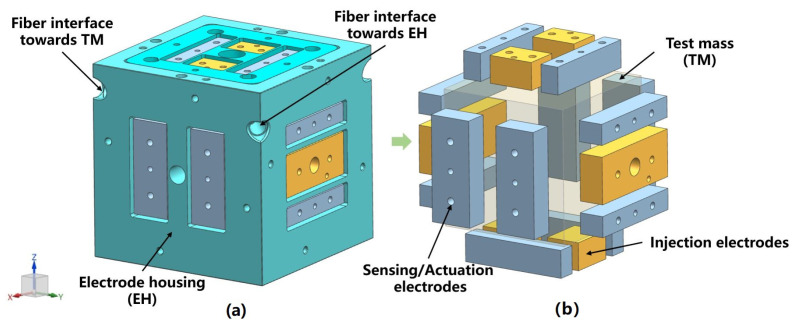
(**a**) Inertial sensor used in LISA and its (**b**) six-degree-of-freedom differential capacitance pair.

**Figure 2 sensors-23-07794-f002:**
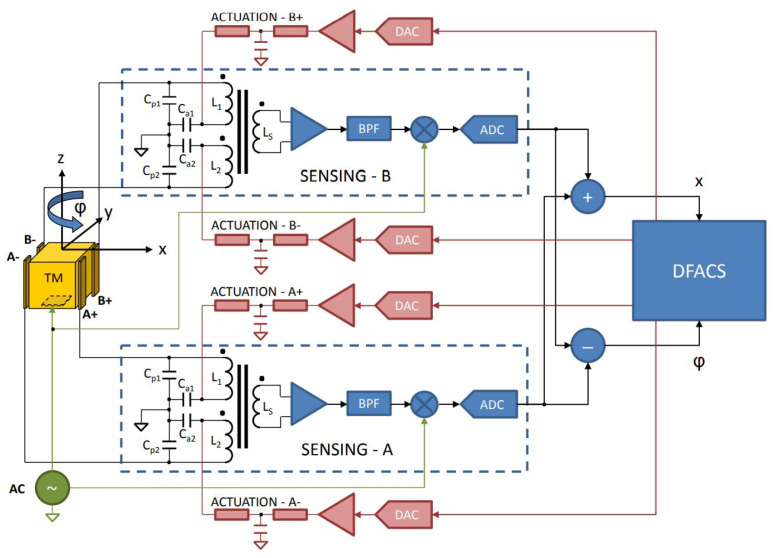
The capacitor sensing and actuation circuit for translation along the *x*-axis and rotation around the *z*-axis. The same scheme is implemented for the remaining four DOFs.

**Figure 3 sensors-23-07794-f003:**
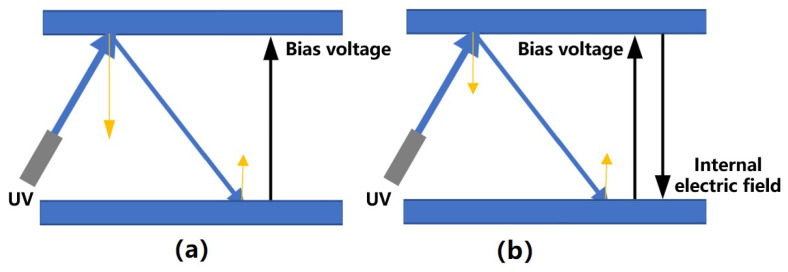
(**a**) UV discharge process controlled by bias voltage and (**b**) its charge balance state, where the yellow arrows represent the direction and magnitude of the photocurrent.

**Figure 4 sensors-23-07794-f004:**
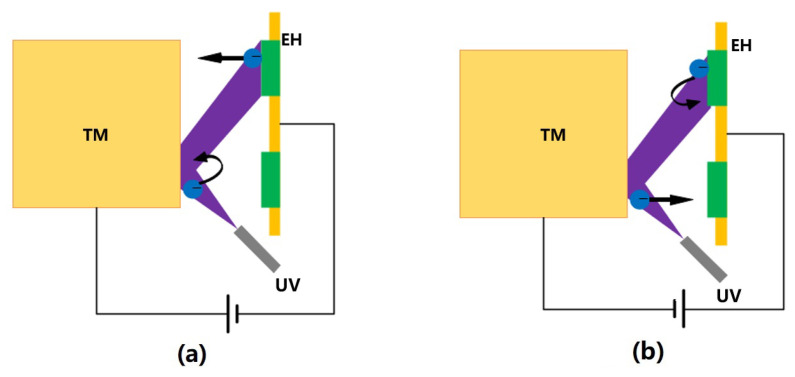
DC discharge diagram: (**a**) positively biased TM, (**b**) negatively biased TM.

**Figure 5 sensors-23-07794-f005:**
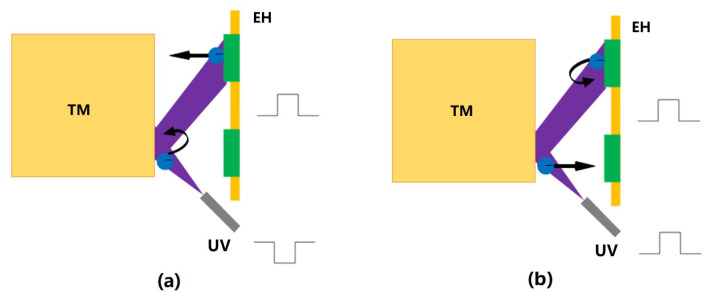
AC discharge diagram: (**a**) the out-of-phase case, (**b**) the in-phase case.

**Figure 6 sensors-23-07794-f006:**
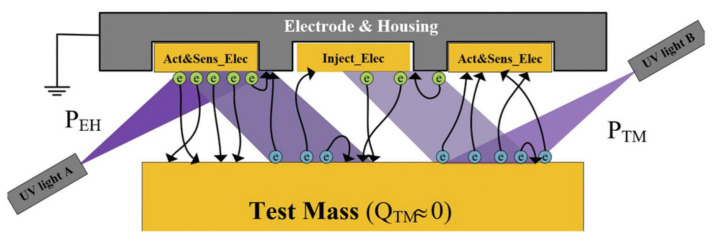
Adaptive discharge method based on differential illumination model.

**Figure 7 sensors-23-07794-f007:**
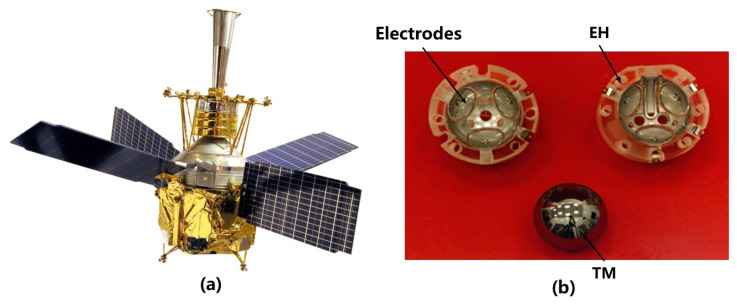
(**a**) The GP-B mission and (**b**) a gyroscope with its electrode housing.

**Figure 8 sensors-23-07794-f008:**
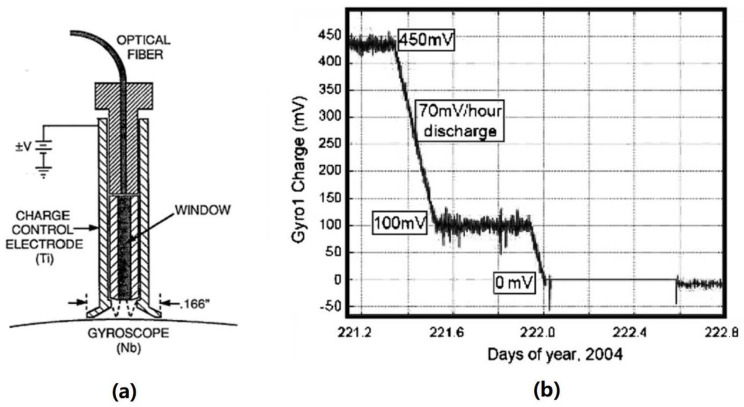
(**a**) Schematic of UV counter electrode and (**b**) part of its on-orbit discharge results.

**Figure 9 sensors-23-07794-f009:**
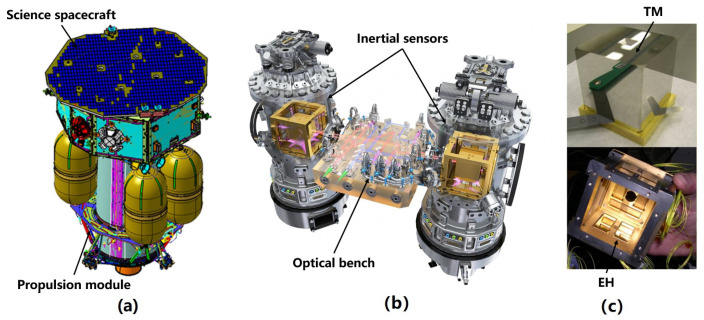
(**a**) The LPF mission, (**b**) the LISA Technology Package it carries, and (**c**) the sensitive probe of an inertial sensor.

**Figure 10 sensors-23-07794-f010:**
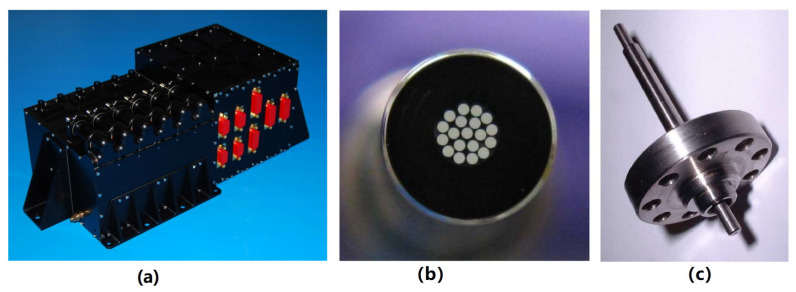
The (**a**) ULU, (**b**) FOH, and (**c**) ISUK of the LPF CMS.

**Figure 11 sensors-23-07794-f011:**
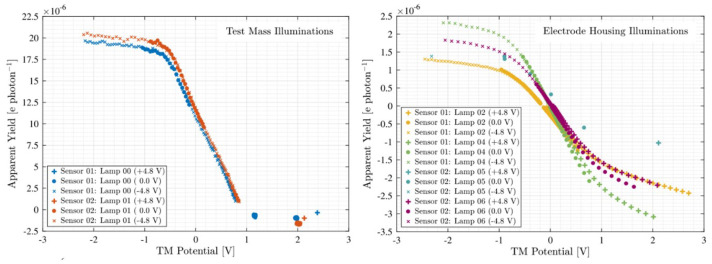
On-orbit discharge curves of LPF CMS.

**Figure 12 sensors-23-07794-f012:**
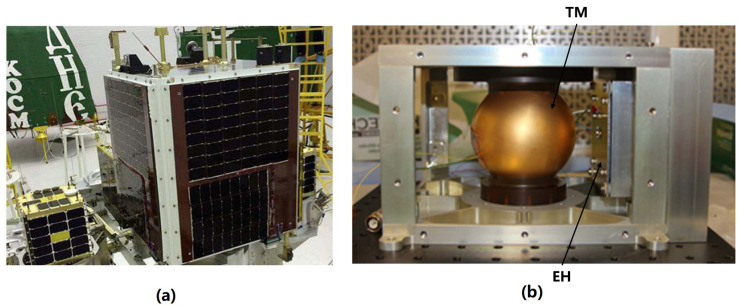
(**a**) The SaudiSat-4 mission and (**b**) its UV LED based CMS.

**Figure 13 sensors-23-07794-f013:**
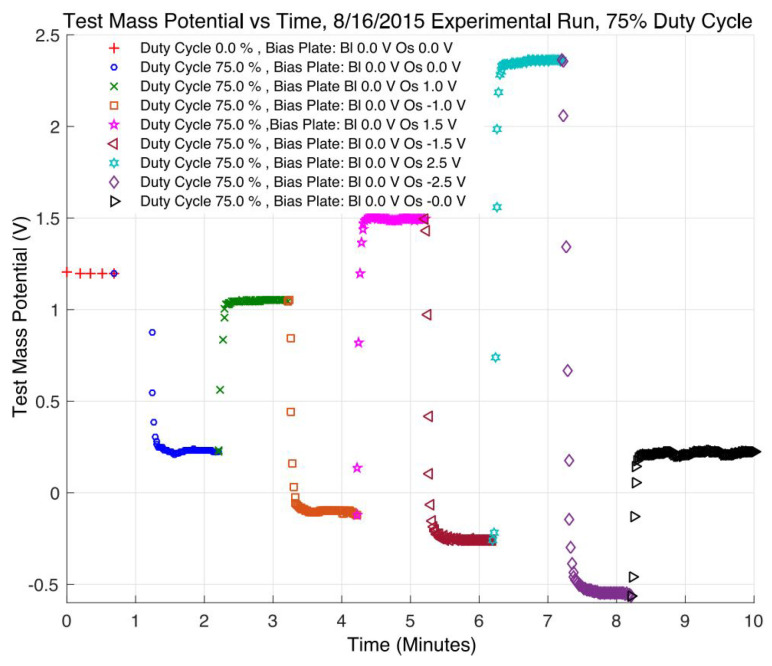
The experimental results of AC discharge at 75% duty cycle.

**Table 1 sensors-23-07794-t001:** Comparison of UV LEDs with conventional mercury lamps.

	UV LEDs	Mercury Lamps
Package	Small size, light weight.	Large size and weight.
Power dissipation	Low power consumption, small heat.	High power consumption and more heat.
Lifetime	10,000–50,000 h.	2000–10,000 h.
Stability	Good stability and can work normally under thermal vacuum, radiation, vibration, and impact.	Sensitive to temperature and adverse conditions.
Operation	Current drive, easy to couple.	Long opening time, awkward to operate.
Environmental impact	Non-toxic harmless.	Contain toxic metals and will produce RFI and EMI.
Modulation performance	Can be modulated at high frequency to achieve high dynamic range in AC CMS.	Not easy to modulate; the dynamic range is limited and cannot be used for AC CMS.

## Data Availability

Not applicable.

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
