# Peer review of "High-Precision Inertial Sensor Charge Management Based on Ultraviolet Discharge: A Comprehensive Review"

_sensors, 2023, doi:10.3390/s23187794_

Round 1

Reviewer 1 Report

This manuscript is a nice review of the technology of inertial mass accelerations used for future high-technology space-based gravitational wave detectors. Some improvements can be offered, but are not required in the present paper:

1) Mutual inductance of the capacitor sensing and activation circuit may lead to unwanted acceleration frequencies and so should be commented on.

2] Eddy currents are present in transit events and a discussion of their significance would be expected.

3] It is not required now, but future reviews should have a glossary of terms at the end (e.g. TM == test mass, etc.).

The only English mistake is in line 35 for the Greek letter mu.

Reviewer 2 Report

See detailed comments as highlights with notes in attached pdf file.

Satisfactory

Reviewer 3 Report

The review paper discusses the challenge of noise caused by charge accumulation from cosmic rays and solar particles in space gravitational wave detection. It highlights the use of ultraviolet discharge for charge management in high-precision inertial sensors, detailing principles, methods, and implementation. The paper explores key considerations for coatings, light sources, and optical paths, offering valuable insights for developing effective charge management systems and advancing ultraviolet discharge technology. This work is of interest, however, there are some concerns of this reviewer to be addressed. Please find below my comments:

1. Given that inertial sensors play a critical role in providing references for precision space missions, could you delve into the specific ways in which the charge accumulation resulting from the space environment affects their on-orbit performance and potential noise interference?

2. Could you provide a detailed explanation of the non-contact UV discharge technology's mechanism for managing charges in high-precision inertial sensors, particularly focusing on how it utilizes the photoelectric effect to effectively remove residual charges from the TM?

3. In the context of meeting the stringent noise requirements for scientific measurements in space gravitational wave detection, can you elaborate on the dynamic range constraints that the related CMS must fulfill and how these constraints relate to the indirect noise effects arising from the discharge process?

4. The conclusion mentions AC discharge as a more promising approach for charge management due to not requiring a special bias voltage. Could you elaborate further on the factors that influence the accuracy and effectiveness of AC discharge, especially in regard to the photoelectric properties of the surface coating and their impact?

5. With the successful implementation of non-contact CMS in various space missions, such as GP-B, LPF, and SaudiSat-4, could you elaborate on the challenges that remain to be addressed for the further development of CMS in the context of future space gravitational wave detection missions? Specifically, what are the complexities involved in measuring and recovering the photoelectric properties of the coating, as well as packaging, controlling the light source, and designing the optical path?

6. The adaptive discharge method is mentioned as a potential solution for addressing challenges related to complex physical parameters and photoelectric properties. Can you provide more details on how this method works, including the process of calibrating the compensation coefficient and how it adapts to changes in physical parameters? Furthermore, could you explain the specific scenarios where the effectiveness of this method needs to be verified?

7.In light of the increasing demands of future space gravitational wave detection missions, could you provide insights into the advancements or breakthroughs required in understanding the physical process behind UV discharge? Additionally, could you elaborate on how these advancements would impact the design and arrangement of optical paths, as well as the control and packaging of the light source, to further enhance the capabilities of non-contact CMS?

Minor edit might required. Overall, it is well-written.

Round 2

Reviewer 2 Report

The authors addressed most comments adequately.

However, as a review article, the references are a major component. Thought comprehensive, the reference list remains lacking.

Suggest thorough review, cleaning, and updating of the reference list. In particular, add the up to date papers, with actual results of the different missions described and with emphasis on CMS data.

Adequate.

Reviewer 3 Report

Thank you for addressing all comments of this reviewer. I have no more comments.

Minor edit required.

Author Response

Thank you for your support and thanks again for your time!